# Highly Porous Co-Al Intermetallic Created by Thermal Explosion Using NaCl as a Space Retainer

**DOI:** 10.3390/ma17174380

**Published:** 2024-09-05

**Authors:** Yonghao Yu, Dapeng Zhou, Lei Qiao, Peizhong Feng, Xueqin Kang, Chunmin Yang

**Affiliations:** 1China University of Mining and Technology, Xuzhou 221116, China; yuyrrow131014@163.com (Y.Y.); fengroad@163.com (P.F.); 2China Coal Technology Engineering Group, Huaibei Blasting Technology Research Institute Limited Company, Huaibei 235000, China; hbzhoudapeng@126.com; 3Xuzhou XCMG Excavator Machinery Co., Ltd., Xuzhou 221000, China; 18852141789@163.com

**Keywords:** porous materials, intermetallic, space retainer, thermal explosion, sintering

## Abstract

Co-Al porous materials were fabricated by thermal explosion (TE) reactions from Co and Al powders in a 1:1 ratio using NaCl as a space retainer. The effects of the NaCl content on the temperature profiles, phase structure, volume change, density, pore distribution and antioxidation behavior were investigated. The results showed that the sintered product of Co and Al powders was solely Co-Al intermetallic, while the final product was Co_4_Al_13_ with an abundant Co phase and minor Co_2_Al_5_ and Co-Al phases after added NaCl dissolved out, due to the high T_ig_ and low T_c_. The open porosity of sintered Co-Al compound was sensibly improved to 79.5% after 80 wt.% of the added NaCl dissolved out. Moreover, porous Co-Al intermetallic exhibited an inherited pore structure, including large pores originating from the dissolution of NaCl and small pores in the matrix caused by volume expansion due to TE reaction. The interconnected large and small pores make the open cellular Co-Al intermetallic suitable for broad application prospects in liquid–gas separation and filtration.

## 1. Introduction

Intermetallics have the common merits of both metals and ceramics, because most of the elements in these materials are bonded through covalent bonds, while a small number of elements are bonded through metal bonds [1,2,3]. Intermetallics have extensive industrial applications, including in the automobile industry and energy production [3,4], due to their distinguished performance characteristics, such as lower density, higher elastic modulus, superior specific strength, machinability, outstanding corrosion resistance and excellent antioxidant capacity under high temperature [3,5,6]. Al-based porous intermetallics have been extensively researched and used in catalytic agents, filtering devices and thermal insulation parts in harsh environments [7,8,9], mainly due to their low densities and high melting temperatures. The technologies of Al-based porous intermetallics focus on Ti-Al [4,6,7,8,10,11,12], Fe-Al [13,14], and Ni-Al [15,16,17]. Due to their advantages of low density, excellent corrosion resistance and low cost, significant progress has been made in the investigation and application of Ti-Al and Fe-Al.

Co-based intermetallics generally have high melting points, improved resistance to oxidation at high temperatures, outstanding hot anticorrosive properties and excellent thermal fatigue resistance; consequently, they have been used for gas turbine blades to reduce their environmental impact [18,19]. Porous Co-Al intermetallics with a high specific surface area and high penetration rate have broad application prospects in industry [20,21]. They can be used in catalytic carriers and filtering materials [22,23]. Yeh et al. [24] prepared a Co-Al intermetallic compound using the SHS method and recorded a highly exothermic reaction in the combustion process. Lekatou et al. [25] showed that Al-32%Co alloys produced by casting, arc melting under argon or powder metallurgy exhibit high resistance in 3.5% NaCl at 25 °C. Kang [26] and Shang [27] used thermal explosion (TE) to fabricate Co-Al porous intermetallics. The prepared Co-Al intermetallics showed outstanding antioxidation properties in air at 650 °C and excellent compressive strength. There are micro- and nano-scale pores in Co-Al intermetallics, but the porosity of these compounds is still relatively low.

Nowadays, various technologies have been developed to prepare high-porosity intermetallics. Sacrificial template, extrusion and space holder processes are often used to produce porous materials. The sacrificial template technique causes pollution to the environment due to the decomposition of polymer templates [28]. Extrusion is mainly used to fabricate materials with directional pores [29]. The space holder process is an environmentally friendly and economical method for controlling pore size and porosity. NaCl, sucrose and urea are common space holders. NaCl is a low-cost, reusable, and environmentally friendly pore-forming agent [30]. Jiao [31] and Wang [32] used NaCl as a space holder to fabricate TiAl_3_ and TiAl porous intermetallics by the TE reaction, with the open porosity reaching 86.3% and 84%, respectively. Research on the effect of pore-forming agents on the porosity of Co-Al materials is limited.

In this work, the TE reaction was used to manufacture Co-Al intermetallic compound porous materials. NaCl particles were used as space retainers to increase the open porosity of the Co-Al intermetallics and were thoroughly dissolved out through a water leaching method before sintering. The influence of NaCl content on the temperature profiles, phase structure, volume change, density, pore distribution and antioxidation behavior is discussed in this paper.

## 2. Experiments

The particle size and purity of Co and Al powders (provided by Wodetai Technology Co., Ltd., Beijing, China) were 18 μm and 99.9% and 47 μm and 99.0%, respectively. Two powders were blended in a 1:1 molar ratio. NaCl with a particle size of 200–500 μm and a purity of 99.5% was used as a temporary space retainer. The entire fabricated operation is demonstrated in Figure 1. Co and Al powders were thoroughly blended according to the method described by Kang in [26]. Subsequently, NaCl granules were mingled with fine, powdery Co-Al material in a ceramic vessel. The mixture in this vessel was manually stirred with a glass rod for 10 min under atmosphere at normal temperature. The green compacts with 50, 60, 70 and 80 wt.% fraction of NaCl were prepared, respectively. A few drops of ethanol were dripped into the Co-Al-NaCl blend to prevent aggregation. The mixture was then placed in a stainless-steel cylinder-shaped die at a uniaxially pressure of 200 MPa at normal temperature to generate a disc-shaped green compact, which had a diameter of 16 mm and thickness of approximately 3 mm. For comparison, the Co-Al sample was prepared according to the same process. The next step was to dislodge NaCl from the green compact. The green compact was placed in water at 70 °C for 30 h to dissolve the NaCl. It was then ensured that NaCl was totally dissolved out based on the quality before and after leaching.

The green compact was sintered in a tube furnace filled with argon gas. The furnace was heated to 700 °C at a rate of 10 °C/min; the temperature was maintained for 30 min and then cooled to normal temperature inside the tube furnace. To examine the TE reaction of Co-Al, a pair of thermocouples, each about 0.1 mm in diameter, was installed between two identical samples to measure the reaction actual temperature.

The sample dimensions before and after sintering were gauged to account for the change in axial direction, radial direction and volume. The Archimedes method was used to measure the density of the Co-Al intermetallics. The open porosity of the Co-Al porous intermetallics was calculated using the following formula: equation *θ* = (*M_a_* − *M*_0_)/*Vρ*, where *M*_0_ is the mass of Co-Al green compact, *M_a_* is the mass of sintered Co-Al porous intermetallic filled with wax, *V* is the external volume of sintered porous product, and *ρ* is wax density.

Powders ground from sintered porous products were identified using X-ray diffraction (XRD). A Cu target (λ = 0.15406 nm) was used under radiation of 40 kV and 150 mA during the XRD testing process. An optical microscope was used to examine the morphology of polished samples. The microtopography of the Co-Al compound fracture surface was checked using scanning electron microscopy (SEM). An oxidative test was carried out in a muffle furnace at 650 °C for 96 h under air atmosphere. Oxidation kinetic curves of Co-Al intermetallics after different amounts of NaCl dissolved out were drawn based on mass changes measured every 24 h.

## 3. Results and Discussion

### 3.1. Exothermic Curves of the Co-Al System

Figure 2 shows the whole exothermic curves and the exothermal peaks of the Co-Al sample and Co-Al samples after different amounts of NaCl were dissolved out during heating in a tube furnace. The TE curves were divided into three parts based on the ignition temperature (T_ig_) and combustion temperature (T_c_) identified in Figure 2. The sample temperature increased slowly in the initial stage according to the furnace setting and then started to climb rapidly from T_ig_ to T_c_, reflecting that the sample was ignited and a large amount of heat was discharged. Afterwards, the measured temperature rapidly dropped to the set furnace temperature and was maintained for 0.5 h. Finally, a Co-Al porous intermetallic was acquired after the furnace cooled to normal temperature. The temperature T_ig_, T_c_, and total reaction time from T_ig_ to T_c_ are the most significant characteristics of the TE reaction. The T_ig_ value of the Co-Al sample was 645.1 °C and below the set furnace temperature (700 °C) or melting point of Al (660 °C), showing that the Co-Al reaction had already begun between solid Co and solid Al [33,34]. The T_ig_ values were 669.4, 669.5, 670.4 and 678.0 °C for specimens with 50, 60, 70 and 80 wt.% NaCl added and dissolved out, respectively. These T_ig_ values were below the set furnace temperature but above the melting temperature of Al, indicating that the Co-Al reaction also occurred in solid Co and liquid Al [32,34]. The ignition temperature increased with the NaCl content because the TE reaction was inhibited by the pores left after the pore-forming agent NaCl dissolved out. These pores caused a decrease in the contact area between Co and Al and then limited the diffusion between them. The T_c_ values in Figure 2 were 1327.2, 903.1, 949.6, 788.3, and 707.5 °C and were higher than the set furnace temperature, implying there was a significant exothermic reaction between Co and Al. The heat release times of these samples are 5, 26, 29, 23 and 42 s. The combustion temperatures T_c_ were different because the ignition temperature, reaction (XRD result in Figure 3) and reaction time were different under different NaCl contents. The smaller the reacted amount of Co and Al in the same volume decreased, the longer the reaction time and the lower T_c_ temperature.

### 3.2. Analysis of TE Reaction Products

Figure 3 shows XRD spectra of TE compacts. In the specimen with Co and Al in a 1:1 ratio, the product was solely Co-Al intermetallic (Figure 3a). When the added NaCl was dissolved out, the XRD spectra of sintered products were consistent. Figure 3b,c show the XRD pattern of the sintered products after 50 and 60 wt.% NaCl was added and dissolved out. The predominant intermetallic was Al_13_Co_4_ with minor abundant Co, Al_5_Co_2_ and Co-Al phases. In the green compact with Co and Al in a 1:1 ratio, a large amount of heat was released during the reaction between Co and Al, so the T_c_ reached its highest value: 1327.2 °C. According to the Co-Al phase diagram, Co and Al can fully react at high temperatures to form Co-Al intermetallics. Co and Al powders came into uniform contact with NaCl when 60 wt.% NaCl was added, but the contact between Co and Al was dispersed. The reaction between Co and Al was inhibited and the heat released during the reaction process decreased after 60 wt.% NaCl was added and dissolved out. The value of T_c_ decreased to 949.6 °C. The diffusion of Al to Co was insufficient, resulting in the formation of Al_5_Co_2_ and Al_13_Co_4_ in the Al powder region and Co-Al at the interface of Co and Al. It is difficult for Al to diffuse into the interior of Co particles, resulting in the presence of Co in sintered Co-Al compacts after 60 wt.% NaCl was added and dissolved. The smaller the amount of reaction, the longer the reaction time, resulting in a lower T_c_. According to the Co-Al phase diagram, when the T_c_ value is too low, it is difficult to form Co-Al intermetallics. Instead, Al_5_Co_2_ and Al_13_Co_4_ may form near the low melting point of Al powder, potentially leaving some Co unreacted.

### 3.3. Analysis of Density and Volume Change

Figure 4a shows the macroscopic appearance of Co-Al compacts and Co-Al samples after added NaCl dissolved out. Both Co-Al green compact and Co-Al green compacts after NaCl dissolution maintained their integrity. Tiny holes can be observed on the green compact surface. After sintering, although all the samples still conserved their disc shape, there was significant volume expansion. Figure 4b shows the volume change with varying NaCl content. The volume change trend is the same as the dimension change in both the axial and radial direction. The volumetric expansion ratio of the Co-Al sample was 12.3%. This value changed to 37.4, 20.9, 7.2 and 4.6% after 50, 60, 70 and 80 wt.% NaCl were added and dissolved out, respectively. In Co-Al green compact, the pores in the sample are not interconnected. During the thermal explosion reaction, the rapidly increasing temperature causes the gas in the pores to expand rapidly, while the compact sample limits its expansion, resulting in a low expansion rate. The compactness of Co-Al green compact decreased and there were many pores after 50 wt.% NaCl was added and dissolved out. The rapidly increasing temperature causes the gas in the pores to expand rapidly during the thermal explosion reaction, and the expansion rate of the sample increases due to the large number of pores, and these pores cannot be smoothly connected to the outside. Although the T_c_ value decreases, the compactness of the sample also decreases, and the limiting force to sample expansion decreases, and the expansion rate of the sample increases. With the increase in NaCl, although the compactness of the sample decreases, the pores are interconnected and fully connected to the outside. At the same time, the T_c_ value further decreases, causing the expansion rate to decrease gradually. Figure 5 shows the density change of the sintered compacts with varying NaCl content. The density of the sintered compacts decreased with the increase in the NaCl addition amount, because NaCl was dissolved out to form pores, and the volume of the sample increased after sintering. The density values were 3.42, 1.41, 1.11, 0.79, and 0.52 g cm^−3^, corresponding to the Co-Al sample and Co-Al specimens after 50, 60, 70, and 80 wt.% NaCl were added and dissolved out, respectively. The addition of NaCl effectively reduced the density of sintered Co-Al compounds.

### 3.4. Microscopic Analysis of Pore Structure

The optical microscopic morphologies of Co-Al compounds with different NaCl contents are shown in Figure 6. The white bright region represents Co-Al compounds, while the black region is the location where the hole exists. The quantity of large pores among the Co-Al matrix was increased with the NaCl content increasing. Small pores on the matrix were mainly due to the capillary force of liquid Al and eventually precipitated from the Co-Al compound [33,35]. The formation of small pores connected the large pores inherited from NaCl and increased the open porosities of the intermetallic. Figure 7 shows SEM fracture morphologies of Co-Al compounds with different NaCl contents. Sintered Co-Al porous intermetallic compounds featured open characteristics, with these pores all interconnected and, in particular, large pores among the skeleton of the Co-Al compound. The connectivity of pores was very significant after the added NaCl was dissolved out. Figure 8 shows the values of open pores in the Co-Al compound matrix with varying NaCl contents. The values of open pores of the Co-Al compound increased alongside the NaCl content increasing, which were approximately 43.2, 69.5, 72.6, 76.5, and 79.5%, corresponding to the Co-Al sample and Co-Al specimens after 50, 60, 70, and 80 wt.% NaCl were added and dissolved out. These increased pores were mainly formed by the dissolution of NaCl.

### 3.5. Analysis of Co-Al Compound Antioxidation Performance

Figure 9 shows the weight gain of the Co-Al compound during oxidation experiments in air atmosphere at 650 °C. All curved lines displayed a parabolic shape, illustrating that Co-Al porous compound has excellent antioxidation at the elevated temperature. The values of weight increase were 1.6, 183.7, 138.8, 126.0 and 61.8 g m^−2^ after 96 h of oxidation experimentation, which corresponded to 0, 50, 60, 70 and 80 wt.% NaCl content, respectively. The specimen without NaCl had the best high-temperature oxidation resistance, due to the product of these specimens being a single phase of Co-Al. During the oxidation process, a dense Al_2_O_3_ protective film forms on the Co-Al phase; this film hinders further oxidation of Co-Al [36]. At the same time, due to the high density and small specific surface area of the Co-Al sample, the weight gain is the smallest in the oxidation test. Dense Al_2_O_3_ film can also be formed on the surfaces of Al_13_Co_5_ and Al_5_Co_2_ phases at 650 °C, but there was a minor abundant Co element when NaCl was added, and this element can be oxidized into cobalt oxide at about 300 °C, so the weight gain of the Co-Al compound after the added NaCl dissolved out was higher than that of the Co-Al compound. The weight gain of the Co-Al porous compound after the added NaCl dissolved out decreased with an increase in the NaCl content, because the amount of Co-Al intermetallic per unit area was reduced and the amount of cobalt oxide decreased. Although the oxidation resistance was reduced, Co-Al still exhibited excellent antioxidation compared with the Ti-Al compound [33].

## 4. Conclusions

(1)High-porosity and low-density Co-Al compounds were manufactured using a high-efficiency and energy-conserving thermal explosion method with a leachable NaCl space retainer. Before sintering, the added NaCl was dissolved out completely.(2)The T_ig_ value of the Co-Al specimen was near to the melting temperature of Al, while the temperatures of the Co-Al specimens after the added NaCl dissolved out were higher than the melting temperature of Al. But the T_c_ of the Co-Al specimen was higher than that of the Co-Al specimens after the added NaCl dissolved out.(3)The final product was solely Co-Al intermetallic for the Co-Al sample, while the final products were Al_13_Co_4_ with minor abundant Co, Al_5_Co_2_ and Co-Al phases with Co-Al specimens after the added NaCl dissolved out, due to the low T_c_.(4)The maximum opening porosity of the sintered Co-Al compound was 79.5% after 80 wt.% NaCl was added and dissolved out. Porous Co-Al compounds displayed an inherited pore structure, including large pores originating from the dissolution of NaCl and small pores in the Co-Al matrix caused by volume expansion owing to TE reaction.(5)The specimen without NaCl exhibited the best high-temperature oxidation resistance because it was a single phase and formed a dense oxide film more easily at high temperatures. High-temperature antioxidation of the Co-Al compound after the added NaCl dissolved out was not high as that of the Co-Al specimen because the abundant Co could be oxidized into cobalt oxide, decreasing its oxidation resistance. In the future, we can increase the oxidation resistance of sintered products by increasing the furnace temperature to produce a single Co-Al intermetallic or reducing the residual amount of Co.

## Figures and Tables

**Figure 1 materials-17-04380-f001:**
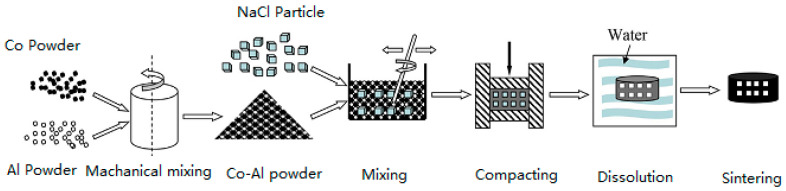
Illustration of the fabricated process.

**Figure 2 materials-17-04380-f002:**
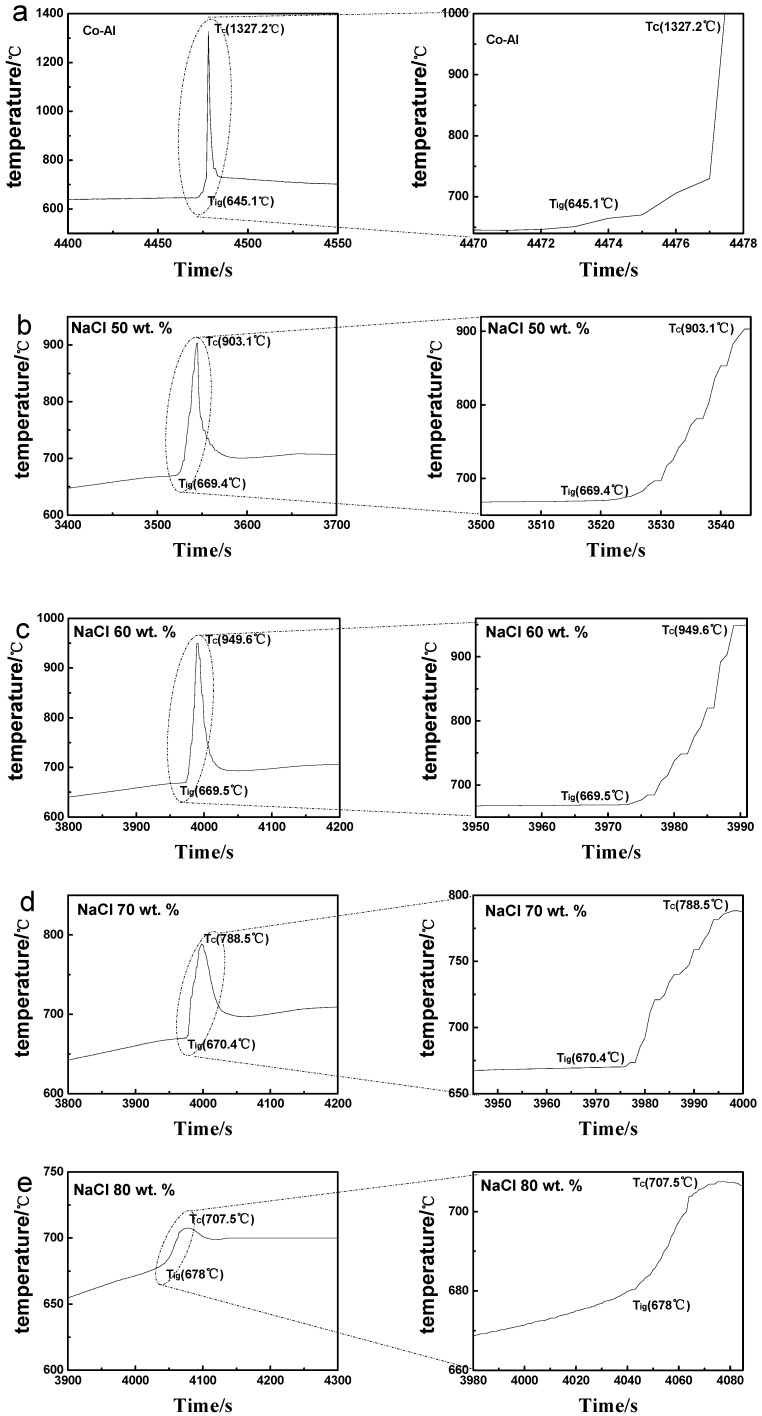
Temperature–time profiles (left) and magnified exothermic curves (right) of (**a**) Co-Al sample and Co-Al samples after (**b**) 50, (**c**) 60, (**d**) 70 and (**e**) 80 wt.% NaCl was added and dissolved out.

**Figure 3 materials-17-04380-f003:**
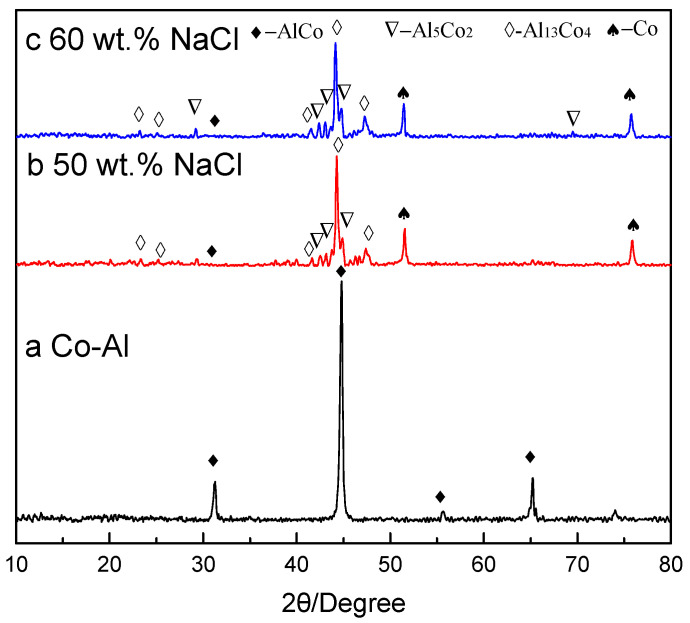
XRD spectra of sintered (**a**) Co-Al sample and Co-Al sample after (**b**) 50 wt.% and (**c**) 60 wt.% NaCl was added and dissolved out.

**Figure 4 materials-17-04380-f004:**
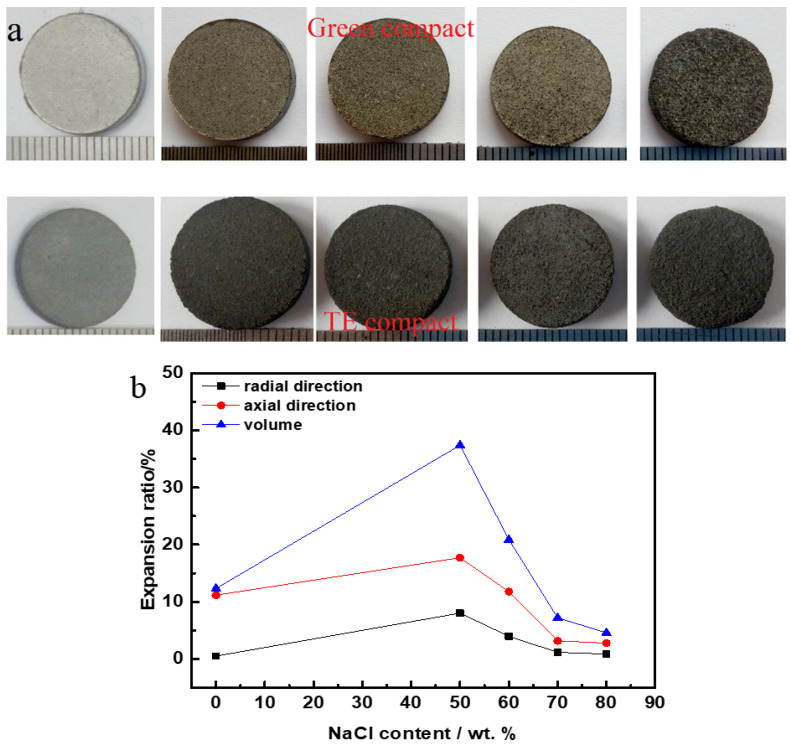
Expansion behaviors of porous intermetallics with different NaCl contents: (**a**) macroscopic, (**b**) volume, axial and radial direction expansion.

**Figure 5 materials-17-04380-f005:**
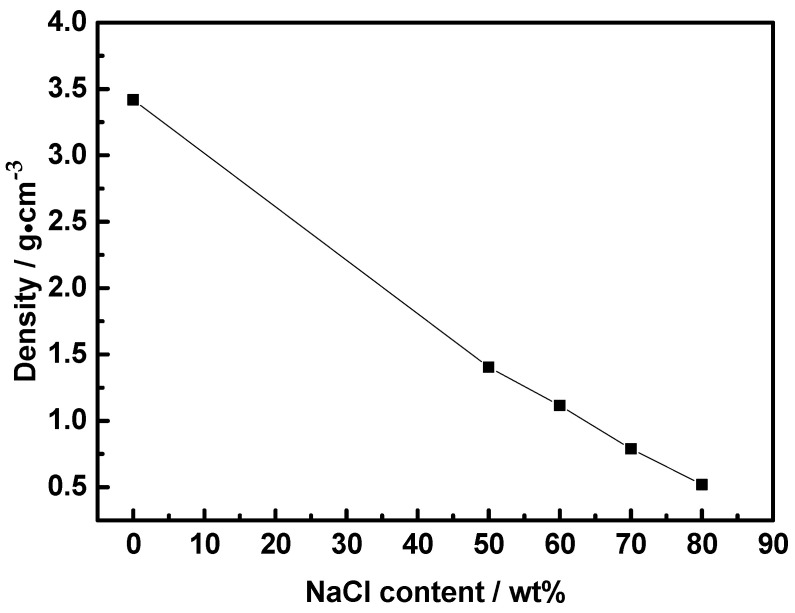
Specimen density after TE reaction with different NaCl contents.

**Figure 6 materials-17-04380-f006:**
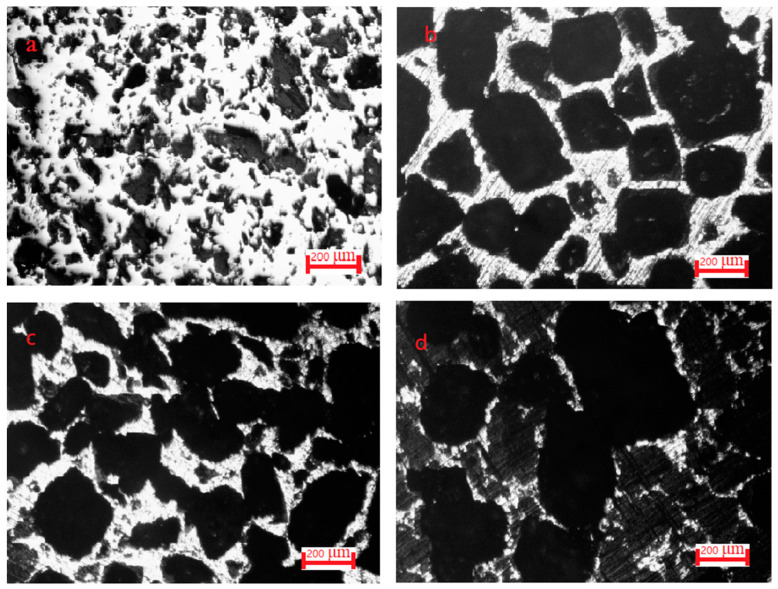
Optical microscope images of polished (**a**) Co-Al specimen and Co-Al specimens after (**b**) 50, (**c**) 60 and (**d**) 80 wt.% NaCl was added and dissolved out.

**Figure 7 materials-17-04380-f007:**
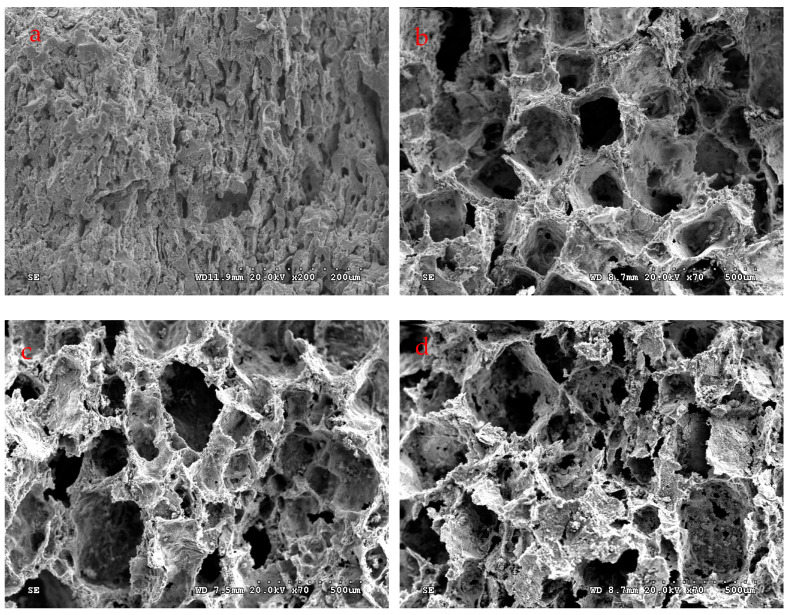
SEM fracture micrographs of sintered (**a**) Co-Al product and Co-Al products after (**b**) 50, (**c**) 60 and (**d**) 80 wt.% NaCl was added and dissolved out.

**Figure 8 materials-17-04380-f008:**
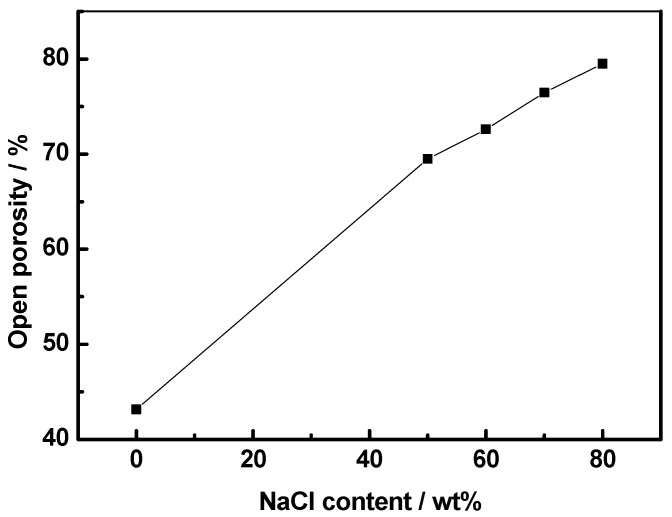
Open porosity of Co-Al porous intermetallic after different contents of NaCl were added and dissolved out.

**Figure 9 materials-17-04380-f009:**
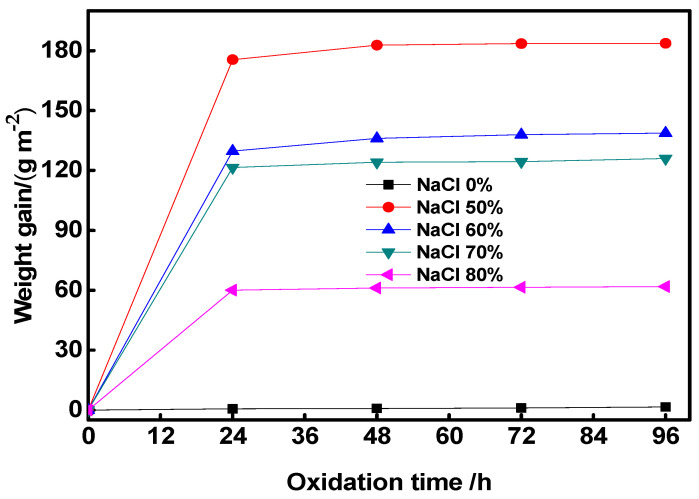
Weight gain of Co-Al porous products after different contents of NaCl were added and dissolved out, oxidized at 650 °C in air.

## Data Availability

The raw data supporting the conclusions of this article will be made available by the authors on request.

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
