# Peer review of "Highly Porous Co-Al Intermetallic Created by Thermal Explosion Using NaCl as a Space Retainer"

_materials, 2024, doi:10.3390/ma17174380_

Round 1

Reviewer 1 Report

Comments and Suggestions for Authors

1) Please explain what are the potential applications for highly porous Co-Al alloys.

2) Introduction needs to be expanded. You may use more relevant works on Co-Al and Al-Co alloys prepared by powder metallurgy:

https://doi.org/10.1016/j.matchar.2023.112785

https://doi.org/10.1016/j.corsci.2012.06.002

3) Did you select NaCl powders with a significantly larger particle size compared to Al and Co powders on purpose? Can the size of NaCl powder be altered to tailor the size of the porous material and/or the shape of pores?

4) Why did you select NaCl as a space retainer?

5) Why when you sinter Co and Al powders without NaCl you get a single phase (Co-Al) while when you add NaCl you get a variety of intermetallics? Please expand the discussion and the mechanism behind this.

6) You need to explain in more detail why the expansion ratio initially increases with NaCl increase and consequently decreases.

7) Figure 6a and figure 7a are exactly the same as in this work: 

https://doi.org/10.1515/htmp-2020-0076

Replace the specific images with different ones.

8) What is the highest percentage of open porosity you may achieve with CoAl alloys before the alloys start collapsing?

9) You need to expand the discussion on the oxidation resistance of the produced porous alloy. You may use relevant works:

https://doi.org/10.3390/ma13143152

Author Response

<Materials> Materials-3151326

< Highly porous Co-Al intermetallic combined by thermal explosion using space retainer NaCl >

Dear Editor,

  Thank you for your useful comments and suggestions on the content and structure of our manuscript. We have modified the manuscript accordingly, and detailed corrections are listed below point by point:

1) Please explain what are the potential applications for highly porous Co-Al alloys.

The potential applications for highly porous Co-Al alloys were added according to some references, and the added content was marked in red. The reference papers were also added in Reference and marked in red.

2) Introduction needs to be expanded. You may use more relevant works on Co-Al and Al-Co alloys prepared by powder metallurgy:

https://doi.org/10.1016/j.matchar.2023.112785

https://doi.org/10.1016/j.corsci.2012.06.002

The first paper is the research work of Xu Chengyi. This paper provides a detailed discussion on the application prospects of Co-Al intermetallic in depression.

The Second paper was written by A. Lekatou. This paper adopts three methods to produce Co-Al materials and tests its corrosion resistance.

These two papers have great reference significance for the preparation and application of Co-Al intermetallic. Therefore, these two papers were cited in the introduction and list as reference in this manuscript.

3) Did you select NaCl powders with a significantly larger particle size compared to Al and Co powders on purpose? Can the size of NaCl powder be altered to tailor the size of the porous material and/or the shape of pores?

The addition of NaCl will affect the external integrity of Co-Al intermetallic. The shape of Co-Al intermetallic can be maintained intact after adding large-sized NaCl, then the integrity of Co-Al intermetallic can also be maintained when adding tailor sized NaCl. Based on the above reason large particle size NaCl were selected as space retainer.

4) Why did you select NaCl as a space retainer?

Common space retainers include NaCl, sucrose and urea. In these space retainers, NaCl is a low-cost pore forming agent compared with other space retainer. It is also a reusable and environmentally friendly pore forming agent compared with urea.

I have added this content in the manuscript and marked in red.

5) Why when you sinter Co and Al powders without NaCl you get a single phase (Co-Al) while when you add NaCl you get a variety of intermetallics? Please expand the discussion and the mechanism behind this.

I have explained and discussed the reaction mechanism of Co and Al in the green compact with Co:Al=1:1. I also explained the reacon for the appearance of Al13Co4, Al5Co2 and Co in sintered Co-Al compact after 60 wt.% NaCl was added and dissolved. All the explanations and discussions were added in the manuscript and marked in red.

6) You need to explain in more detail why the expansion ratio initially increases with NaCl increase and consequently decreases.

I have explained the expansion ration change in the manuscript and marked in red.

7) Figure 6a and figure 7a are exactly the same as in this work: 

https://doi.org/10.1515/htmp-2020-0076

Replace the specific images with different ones.

Figure 6a and figure 7a are exactly the same images with the paper published in HTMP, because they are the same experiment conducted by our research group. We have replaced the specific images of different regions of the same sample due to copyright issues.

8) What is the highest percentage of open porosity you may achieve with CoAl alloys before the alloys start collapsing?

This is a guiding question and this will guide us to add 10~40 wt.% NaCl in subsequent research to study the volume change of Co-Al intermetallic, and obtain the maximum value of volume expansion. This manuscript aims to investigate Co-Al intermetallic can still maintain integrity and has good oxidation resistance even with an increase in porosity after NaCl was added and dissolved.

9) You need to expand the discussion on the oxidation resistance of the produced porous alloy. You may use relevant works:

https://doi.org/10.3390/ma13143152

This paper is very useful to our study. I analyzed the oxidation mechanism of Co-Al intermetallic based on this paper. I also list this paper as a reference. All the revised content was marked in red in the manuscript.

The manuscript has been resubmitted to your journal. We look forward to your positive response.

Sincerely,

Xueqin Kang

Reviewer 2 Report

Comments and Suggestions for Authors

The authors showed an exciting study regarding obtaining porous samples of intermetallic compounds based on the Co-Al system. Please consider the following suggestions and comments.

1)      The authors refer to an experimental procedure (Ref. 25) wherein the metallic powders are ball milled using a high-agency ball mill previous to incorporating the pore former. In this regard, this stage is described as a simple mechanical mixing or blending step. However, strictly speaking, this is not correct from the point of view of the process itself. Ball milling may produce mechanical activation, reactions, amorphization, and alloying, among other processes. Therefore, it is suggested that at least a basic analysis of the ball-milled powders be completed. In this case, it seems that the ball milling stage could not be a simple mixing step but a significant step for further reaction and obtaining the intermetallic compound. Actually, it can also influence the ignition and reaction temperatures.

2)      Figure 3 is incomplete. It is important for the reader to have all of the XRD data available to elucidate the effect of the experimental conditions on the compound's composition.

3)      Labels in Fig. 4b must be revised. It is written "radical direction".

4)      As NaCl is used as a pore former, it is suggested that the  Vol% be included besides wt%.

5)      For the results shown in Fig. 2. Please indicate if the sample size (g or mg) was the same for each experiment.

6)      What is the reason for the differences in color exhibited by the samples?

7)      How is the oper pore volume calculated? The experimental section describes the use of the Archimedes method; however, the SEM analyses reveal macro porous that are even bigger than 200 microns. Is the sample able to retain the water inside this porosity?

8)      Regarding oxidation resistance, authors must consider that besides composition, the different samples also exhibit different surface area values as a result of the obtained porosity.

Author Response

<Materials> Materials-3151326

< Highly porous Co-Al intermetallic combined by thermal explosion using space retainer NaCl >

Dear Editor,

  Thank you for your useful comments and suggestions on the content and structure of our manuscript. We have modified the manuscript accordingly, and detailed corrections are listed below point by point:

 1)      The authors refer to an experimental procedure (Ref. 25) wherein the metallic powders are ball milled using a high-agency ball mill previous to incorporating the pore former. In this regard, this stage is described as a simple mechanical mixing or blending step. However, strictly speaking, this is not correct from the point of view of the process itself. Ball milling may produce mechanical activation, reactions, amorphization, and alloying, among other processes. Therefore, it is suggested that at least a basic analysis of the ball-milled powders be completed. In this case, it seems that the ball milling stage could not be a simple mixing step but a significant step for further reaction and obtaining the intermetallic compound. Actually, it can also influence the ignition and reaction temperatures.

 The reviewer’s description and analysis of the experiment procedure, it will generate mechanical activation, reactions, amorphization and alloying processes. Some papers (<Effect of ball milling time on Fe2O3/Al nano-composite powder>, <Effects of milling time and testing conditions on particle size analysis of alumina powders>) have provided detailed discussions and analyses on this process.

This manuscript mainly studies the influence of thermal explosion reaction on the products and the antioxidant properties of the products, just ensure the consistency of the reactants before the thermal explosion reaction. All reactants Co and Al were ball milling to ensure their consistency. There is no need to explore the changes during the ball milling process.

Of course, this suggestion provides direction for our future research. We can research the influence of Co and Al particle size and fusion process changes on the thermal explosion reaction temperature during ball milling.

2)      Figure 3 is incomplete. It is important for the reader to have all of the XRD data available to elucidate the effect of the experimental conditions on the compound's composition.

 The XRD pattern of the sintered products were basically consistent when added 50, 60, 70 and 80 wt.% NaCl, so only the XRD pattern of sintered product added 60 wt.% NaCl was shown in the manuscript. To demonstrate the consistency of the sintered product added NaCl, another XRD pattern with 50 wt.% NaCl was added.

3)      Labels in Fig. 4b must be revised. It is written "radical direction".

 I have revised “radical direction” to “radial direction”.

4)      As NaCl is used as a pore former, it is suggested that the  Vol% be included besides wt%.

 Porosity is calculated based on the volume ratio to porous materials, and it is the best to measure each reactant according to the volume ration in the manuscript. However, the mixing materials were NaCl, Co and Al powders, their volume and density are difficult to measure, so mass ratio was used for measurement in this manuscript. It also can reflect the influence of NaCl content on porosity of Co-Al intermetallic.

5)      For the results shown in Fig. 2. Please indicate if the sample size (g or mg) was the same for each experiment.

 The size of all samples is a sheet with a diameter of 16mm and a thickness of approximately 3mm.

6)      What is the reason for the differences in color exhibited by the samples?

 There are two reasons for the color difference. One is that slight corrosion occurred during the dissolution process of added NaCl, because the green compact was placed in water at 70 ℃ for 30h to dissolve the added NaCl. Another reason is due to the lighting during the shooting process, because there are significant differences in color for the same ruler in different photographs. We will carefully observe this phenomenon and pay attention to the lighting during the shooting process.

7)      How is the open pore volume calculated? The experimental section describes the use of the Archimedes method; however, the SEM analyses reveal macro porous that are even bigger than 200 microns. Is the sample able to retain the water inside this porosity?

 This is a very instructive suggestion. The sintered products have large macroscopic pores when added NaCl was dissolved out. When using Archimedes’ method to test the porosity, the surface pores may not be able to retain water, but the proportion of surface pores in the sintered product pores is relatively small, and has little impact to the test results. We think that this method can reflect the trend of porosity changing with the amount of added NaCl. This suggestion also prompts us to find better and more accurate method to measure the porosity of materials with macro pores in the following research.

8)      Regarding oxidation resistance, authors must consider that besides composition, the different samples also exhibit different surface area values as a result of the obtained porosity.

 This suggestion is very helpful to us. The mass gain is related to many factors such as sample oxidation, weight, specific surface area, and composition. Some references were used to explain the mass gain obtained from different Co-Al intermetallic. We will consider each factor and study the oxidation performance of Co-Al compounds in the future.

The manuscript has been resubmitted to your journal. We look forward to your positive response.

Sincerely,

Xueqin Kang

Round 2

Reviewer 1 Report

Comments and Suggestions for Authors

Authors have improved the manuscript following the suggestions of the reviewers. This paper is now ready for publication.

Reviewer 2 Report

Comments and Suggestions for Authors

I have carefully revised the revised version of the manuscript. The authors addressed most of the comments from the review and explained some other observations.

Based on the above, I suggest publishing the revised version of the manuscript.